# Spectral Sheaf Filtering: A Topological Approach to Spatio-Temporal Modeling

## Abstract

Spatio-temporal data pose significant challenges for graph-based learning due to their complex, non-stationary dependencies and the limitations of conventional message passing in capturing high-order, asymmetric interactions. We introduce Spectral Sheaf Filtering (SSF), a novel and theoretically grounded framework that redefines information propagation on graphs using the algebraic topology of cellular sheaves. By assigning vector spaces and restriction maps to nodes and edges, SSF encodes context-dependent, localized dynamics that extend far beyond traditional adjacency structures. To further enhance expressivity and efficiency, we introduce spectral filtering over the sheaf Laplacian, enabling frequency-aware decomposition via the graph Fourier transform while emphasizing latent spectral features. This spectral view allows SSF to adaptively modulate information flow across frequency components, effectively mitigating oversmoothing in deep graph neural networks. Extensive experiments on diverse spatio-temporal traffic forecasting benchmarks show that SSF outperforms state-of-the-art methods, especially in long-horizon forecasting tasks. Our results highlight the value of topological structures in advancing graph learning for spatio-temporal systems. The code is available at: `https://github.com/anonymous-submisssion/SSF`.

## 1 Introduction

Urban mobility systems often exhibit remarkably heterogeneous and non-intuitive responses to localized disruptions. For instance, consider a highway accident during rush hour that triggers a cascade of unexpected traffic congestion, propagating across distant neighborhoods while adjacent streets, geographically just as close, remain unaffected, Figure 1. This scenario underscores a core challenge in spatio-temporal modeling: real-world phenomena rarely conform to simple, proximity-based interaction patterns. Instead, they exhibit complex, higher-order dependencies, where the flow of information varies significantly across both spatial and temporal dimensions. This illustrates the need for models that capture non-local, asymmetric, and higher-order interactions.

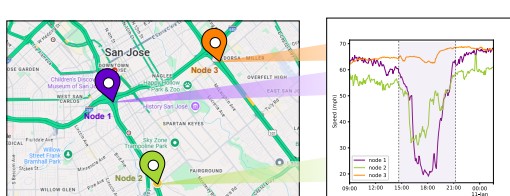

Figure 1: Spatio-temporal traffic dynamics across three sensor nodes in the PEMS-BAY dataset Li et al. (2018). Node 2 (green) exhibits a strong temporal correlation with the significant speed drop at Node 1 (purple), whereas Node 3 (orange), despite similar spatial proximity, remains unaffected.

Recent advances in spatio-temporal modeling have used graph neural networks (GNNs) Sahili & Awad (2023); Wu et al. (2020), attention mechanisms Yu et al. (2024); Feng et al. (2023); Jyotishi & Dandapat (2023), and diffusion-based approaches Yang et al. (2024). However, these methods primarily focus on direct relationships between adjacent locations (first-order spatial relationships) Sahili & Awad (2023) or oversimplified temporal dependencies Yu et al. (2024). Graph-based models, in particular, frequently assume uniform information propagation along edges Sahili & Awad (2023), overlooking dynamic variations in interaction strength. As a result, they often suffer from oversmoothing and fail to capture the intricate, high-order, non-local dependencies inherent in complex systems.

To overcome these limitations, we propose a novel framework that moves beyond conventional spatial adjacency and captures richer topological signals. Our approach draws from differential geometry to model non-linear, high-order relationships via sheaves, offering a significantly more expressive representation than traditional graph structures. To the best of our knowledge, this is the first framework that models spatio-temporal data with graph cellular sheaves. Furthermore, we introduce a spectral filtering technique based on the sheaf Laplacian, which bridges spectral graph theory with sheaf-based graph networks, enabling principled learning on complex spatio-temporal graphs.

A key innovation of our approach lies in utilizing restriction maps from sheaf theory to model data dependencies. These maps enable locally adaptive information propagation, representing a crucial improvement over the uniform message-passing mechanisms in conventional GNNs. By learning how features should transform differently across various regions of the graph, restriction maps enable the model to preserve fine-grained spatial details and avoid issues such as oversmoothing, a common problem that leads to poor performance in tasks requiring localized patterns. Additionally, unlike standard graph Laplacians, the sheaf Laplacian encodes multidimensional interactions through its cellular structure. This allows for richer modeling of signal propagation across spatio-temporal graphs, accommodating dependencies that span beyond simple node-to-node connections.

Afterwards, we perform a spectral decomposition of the sheaf Laplacian, which uncovers the fundamental structure of dependencies in the data and reveals latent topological features encoded in the graph. This decomposition facilitates interpretable and efficient spectral filtering, allowing us to extract and manipulate frequency components of signals defined over the sheaf structure.

While our proposed framework offers general modeling of spatio-temporal data, we specifically demonstrate its effectiveness in the context of traffic forecasting. We conduct extensive experiments on five widely-used benchmark spatio-temporal traffic datasets (METR-LA, PEMS-BAY, PEMS04, PEMS08, NAVER-Seoul), evaluating the model across various prediction horizons. Our approach outperforms state-of-the-art methods, highlighting its strong potential for real-world deployment in transportation systems and other spatio-temporal forecasting applications. In conclusion, our contributions can be summarized as follows:

- This work is the first to model spatio-temporal data using cellular sheaves, effectively addressing the fundamental challenges of traditional GNNs such as oversmoothing and limited expressiveness by capturing the complex, high-order relational structures inherent in such data.

- We introduce a spectral filtering framework applied to the sheaf Laplacian to capture fine-grained graph signal features in the frequency domain effectively.

- Through extensive experiments on spatio-temporal traffic datasets, our framework outperforms state-of-the-art methods, establishing a new benchmark for spatio-temporal prediction performance.

## 2 RELATED WORK

**Spatio-temporal traffic forecasting.** Recent advances in spatio-temporal traffic forecasting have shifted from traditional RNN and CNN models to graph-based architectures to capture the complex spatio-temporal dependencies Bai et al. (2020); Ye et al. (2021). Early work introduced DCRNN Li et al. (2018), which modeled traffic as a diffusion process on a directed graph. This was followed by STGCN Yu et al. (2018), combining graph convolutions with 1D convolutions for temporal patterns. ASTGCN Guo et al. (2019) further advanced data modeling by incorporating attention mechanisms to capture dynamic spatial-temporal correlations. More recently, GMAN Zheng et al. (2020) was proposed, combining spatial and temporal attention mechanisms with a gated fusion module to dynamically model spatio-temporal dependencies in traffic data. Additionally, GMAN proposes a transform attention mechanism that directly links historical and future time steps. Current state-of-the-art approaches include STG-NCDE Choi et al. (2022), which proposes neural controlled differential equations for continuous-time modeling, and DSTAGNN Lan et al. (2022) that integrates dynamic spatial-temporal attention with graph neural networks. Most recently, MegaCRN Jiang et al. (2023b) explicitly disentangles spatial and temporal heterogeneity in traffic data by generating adaptive, context-aware node embeddings using a meta-node bank and hyper-network. Despite these advances, existing methods still struggle to effectively capture long-range and high-order relations that mimic real-world data behavior.

**Sheaf graph neural networks.** Recent advances in GNNs have expanded beyond traditional spatial representations to capture higher-order interactions. Sheaf neural networks represent a promising direction that applies concepts of algebraic topology to model asymmetric data relationships Hansen & Gebhart (2020); Bodnar et al. (2022). Building on cellular sheaf theory, SheafANs Barbero et al. (2022) developed a sheaf attention mechanism that generalizes graph attention networks by integrating cellular sheaves for richer geometric inductive biases. This approach addresses the GNN limitations, oversmoothing, and poor performance on heterophilic graphs, by using transport matrices and sheaf-based feature aggregation to preserve local heterogeneity and geometric structure. In Duta et al. (2023), the authors introduce cellular sheaves for hypergraphs and propose the linear and non-linear sheaf hypergraph Laplacians, generalizing standard hypergraph Laplacians. Our work employs the sheaf theory for modeling the asymmetric high-order dependencies of spatio-temporal data.

**Graph spectral filtering.** Graph spectral filtering is a foundational technique in graph signal processing that operates on the graph's frequency domain, working on the eigenvalues and eigenvectors of the graph Laplacian by emphasizing or attenuating specific frequency components Shuman et al. (2013), Sandryhaila & Moura (2013). The authors in Defferrard et al. (2016) introduced a spectral graph theoretical formulation of CNNs that enables the design of fast, strictly localized filters on arbitrary graph structures. While StemGNN Cao et al. (2020) jointly captures intra-series temporal patterns and inter-series correlations in the spectral domain by integrating Discrete Fourier Transform (DFT) and Graph Fourier Transform (GFT). It also acts as a data-driven learning approach of inter-series relationships without relying on pre-defined topologies, enabling the model to automatically infer graph structures that are interpretable and often superior to manually designed ones. While $S^2$GNNs Geisler et al. (2024) integrates spatial message passing with spectral-domain filters and provides free-of-cost positional encodings, significantly expanding the expressivity and design space of GNN architectures. Specformer Bo et al. (2023a) introduces a transformer-based spectral GNN that uses self-attention to design a learnable spectral filter, capturing both the magnitudes and relative differences of graph Laplacian eigenvalues.

## 3 SPECTRAL SHEAF FILTERING (SSF)

The proposed SSF framework, illustrated in Figure 2, consists of the following. For a graph with $N$ nodes, the model receives its historical observations over the past $T$ time steps, where each time step contains $F$ features (traffic speed), resulting in an input tensor of shape $(N, T \times F)$. These past $T$ temporal frames are concatenated into a single feature vector per node. So, time is treated as part of the input feature dimension. This representation is then passed through an initial MLP that projects it into a higher-dimensional embedding space of size $d \cdot h$, where $d$ is the sheaf stalk dimension and $h$ is the hidden layer size. The spatial dependencies are captured through the sheaf Laplacian. Then at each layer, the model performs a graph Fourier transform with respect to the eigenvectors of the sheaf Laplacian, applies a heat-kernel, and transforms the signal back to the spatial domain. The final node representations are collapsed back and mapped through a linear layer to construct the output predictions of shape $(N, T' \cdot F)$, which are the forecasts for the next $T'$ future time steps. Training is performed using the MSE loss between the predicted and ground-truth future observations.

### 3.1 PRELIMINARIES

Real-world spatiotemporal phenomena, such as traffic density distributions, are inherently embedded within two interconnected mathematical structures: a spatial manifold $\mathcal{M}$ (representing the physical network topology) and a temporal domain $\mathcal{T} \subset \mathbb{R}^+$ (capturing the sequential evolution of measurements). The fundamental challenge in spatiotemporal forecasting lies in modeling the simultaneous interaction between these structures, which traditional GNNs fail to capture adequately due to their tendency to homogenize higher-order and feature-dependent relationships into oversimplified graph representations. To address this, we formulate the problem within the framework of cellular sheaf theory over graphs, processed through spectral analysis.

**Spatial Graph.** Let $\mathcal{G} = (\mathcal{V}, \mathcal{E})$ be a connected, undirected graph representing spatial relationships, where: $\mathcal{V} = \{v_1, v_2, \ldots, v_N\}$ is the vertex set with $|\mathcal{V}| = N$, representing spatial entities (sensors) and $\mathcal{E} \subseteq \mathcal{V} \times \mathcal{V}$ is the edge set modeling spatial connectivity. The adjacency matrix $A \in \{0, 1\}^{N \times N}$ satisfies $A_{ij} = 1$ if $(v_i, v_j) \in \mathcal{E}$, and $A_{ij} = 0$ otherwise.

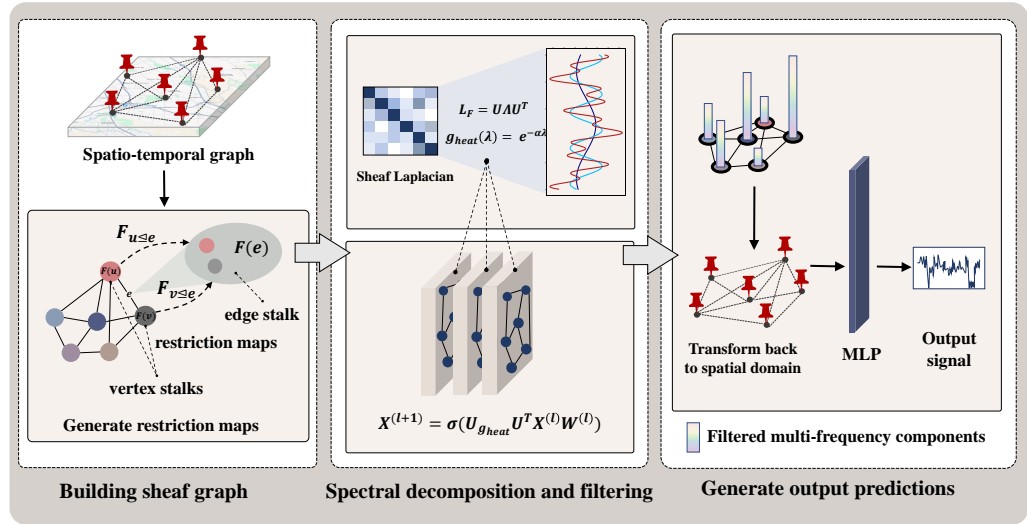

Figure 2: Overview of SSF framework. Spatio-temporal signals are encoded as graph-structured data, followed by sheaf construction assigning vector stalks to vertices and edges. Sheaf Laplacian is then decomposed to enable spectral processing via graph Fourier transform and heat kernel filtering. Finally, the output signal predictions are generated through the last MLP.

**Spatio-temporal Signal.** A spatiotemporal signal is defined as a tensor $X \in \mathbb{R}^{N \times T \times d}$, where: $T$ represents the temporal horizon length, $d$ denotes the feature dimensionality at each node, and each temporal slice $X_t \in \mathbb{R}^{N \times d}$ represents the graph signal at time $t$, with $[X_t]_i \in \mathbb{R}^d$ being the feature vector at node $v_i$ at time $t$.

## 3.2 SHEAF BUILDING

To model higher-order, feature-dependent interactions within the graph, we represent the graph $\mathcal{G}$ with a cellular sheaf $\mathcal{F}$.

**Definition 1 (Cellular Sheaf on Graphs).** A cellular sheaf $\mathcal{F}$ on $\mathcal{G} = (\mathcal{V}, \mathcal{E})$ assigns:

- **Vertex Stalks:** For each vertex $v \in \mathcal{V}$, a finite-dimensional vector space $\mathcal{F}(v) \cong \mathbb{R}^d$.

- **Edge Stalks:** For each edge $e = (u, v) \in \mathcal{E}$, a finite-dimensional vector space $\mathcal{F}(e) \cong \mathbb{R}^d$.

- **Restriction Maps:** For each incidence $v \triangleleft e$, where vertex $v$ is incident to edge $e$, a linear restriction map $\mathcal{F}_{v \triangleleft e} : \mathcal{F}(v) \to \mathcal{F}(e)$, which governs how information from the vertex is projected onto the edge.

The restriction maps encode how node-level features are projected into edge-level representations. We parameterize each $\mathcal{F}_{v \triangleleft e}$ as a learnable linear transformation $\mathcal{F}_{v \triangleleft e} \in \mathbb{R}^{d \times d}$

An important hyperparameter in this setup is the stalk dimension $d$, which controls the capacity of the vector spaces. Higher-dimensional stalks support richer feature propagation and more informative graph signal encoding. When $d = 1$ and $\mathcal{F}_{v \triangleleft e} = 1$, we recover scalar-weighted message passing as in classical GCNs. We empirically analyze the impact of stalk dimensionality in the experiments section. We assume a uniform stalk dimension. All vertex and edge stalks have identical dimension $d$, ensuring that $\mathcal{F}(v) \cong \mathcal{F}(e) \cong \mathbb{R}^d$ for all $v \in \mathcal{V}$ and $e \in \mathcal{E}$.

**Definition 2 (Sheaf Laplacian).** The sheaf Laplacian $L_\mathcal{F}$ is defined as:

$$(L_\mathcal{F} x)_v = \sum_{e=(u,v)\in\mathcal{E}} \mathcal{F}_{v \triangleleft e}^\top \big( \mathcal{F}_{v \triangleleft e} x_v - \mathcal{F}_{u \triangleleft e} x_u \big). \tag{1}$$

We apply the diffusion process using the sheaf Laplacian to minimize the Dirichlet energy function Duta et al. (2023):

$$E_{L_2}^{\mathcal{F}}(x) \;=\; \frac{1}{2} \sum_e \left\| F_{v \triangleleft e} D_v^{-\frac{1}{2}} x_v - F_{u \triangleleft e} D_u^{-\frac{1}{2}} x_u \right\|_2^2 \tag{2}$$

where $D_v = \sum_{e;v \in e} F_{v \triangleleft e}^\top F_{v \triangleleft e}$, and $D = \mathrm{diag}(D_1, D_2, \ldots, D_n)$ is the corresponding block-diagonal matrix. Thus, the sheaf Laplacian $L_{\mathcal{F}}$ models the difference in sheaf-projected features across each edge and aggregates them in a node-specific manner. As such, it captures both structural and semantic discrepancies between nodes in a way that respects the underlying geometry defined by the sheaf. In contrast to using the normalized graph Laplacian which can lead to similar node representation during the diffusion process, using the sheaf Laplacian mitigates over-smoothing. When $d = 1$ and $\mathcal{F}_{v \triangleleft e} = 1$, $L_{\mathcal{F}}$ reduces to the combinatorial graph Laplacian.

### 3.3 Sheaf Fourier Analysis

A common challenge of existing graph neural networks is oversmoothing, where node representations become increasingly similar and eventually indistinguishable as more layers are added. This occurs because repeated message passing causes features of neighboring nodes to converge, ultimately degrading the model's ability to capture discriminative information.

To address this issue, we apply a spectral filtering approach. Spatial-based methods typically propagate information incrementally across layers, limiting each node's receptive field to a bounded local neighborhood Bo et al. (2023b). In contrast, spectral methods leverage the graph Fourier transform, representing node signals as linear combinations of the graph's eigenvectors. This enables the model to access and utilize global structural information from the entire graph in a principled manner. To this end, we extend spectral graph theory to the sheaf-based setting by introducing the spectral decomposition of the sheaf Laplacian.

**Theorem 1 (Sheaf Laplacian Eigendecomposition).** The sheaf Laplacian admits the eigendecomposition:

$$L_{\mathcal{F}} = U \Lambda U^T \tag{3}$$

where $\Lambda = \mathrm{diag}(\lambda_1, \lambda_2, \ldots, \lambda_{Nd})$ and $U = [u_1, u_2, \ldots, u_{Nd}]$ contains orthonormal eigenvectors. The sheaf Laplacian $L_{\mathcal{F}}$ is symmetric positive semidefinite when: the base graph is undirected, edge stalks have inner-product structures, and all restriction maps satisfy $F_{v \triangleleft e}^\top F_{v \triangleleft e} \in \mathcal{S}_d^+$. Also, incidence contributions are accumulated symmetrically. The eigenspace corresponds to the sheaf harmonic sections. The eigenvalues $\lambda$ of the sheaf Laplacian quantify the frequency content of the graph signals and characterize the smoothness of the associated eigenvectors. Small eigenvalues correspond to low-frequency components, which vary slowly across connected nodes and capture global, smooth patterns. In contrast, large eigenvalues represent high-frequency components, which are localized and oscillatory, capturing sharp or irregular changes in the graph signal. The eigenvectors $\{u_\ell\}$ form an orthonormal basis for the space of sheaf signals, thereby generalizing the concept of Fourier modes to the sheaf-theoretic setting. This formulation allows us to design learnable spectral filters that operate over the spectrum of the sheaf Laplacian, enabling the model to selectively enhance or suppress structural features of interest while mitigating the oversmoothing effect.

**Spectral Filtering via Heat Kernel.** Building on the spectral decomposition of the sheaf Laplacian, we apply a heat kernel spectral filter to selectively control the contribution of different frequency components in the graph signal. The motivation behind this choice lies in the need to suppress high-frequency noise and emphasize the low-frequency components that encode the most coherent and structurally meaningful information across the graph.

The heat kernel is defined as:

$$g_{\text{heat}}(\lambda) = e^{-\alpha \lambda}, \quad \alpha > 0 \tag{4}$$

The corresponding diagonal spectral filter matrix is expressed as:

$$\hat{g}_{\text{heat}} = \mathrm{diag}\left(e^{-\alpha \lambda_1}, \ldots, e^{-\alpha \lambda_{Nd}}\right) \tag{5}$$

This filter is then applied in the spectral domain to capture better representation of graph signals while preserving structural patterns encoded in the low-frequency spectrum. Afterwards, node features are

Table 1: Performance on METR-LA, PEMS-BAY, and NAVER-SEOUL Datasets

| Data | Model | 15min / horizon 3 | | | 30min / horizon 6 | | | 60min / horizon 12 | | |
|---|---|---|---|---|---|---|---|---|---|---|
| | | MAE | RMSE | MAPE | MAE | RMSE | MAPE | MAE | RMSE | MAPE |
| METR-LA | ARIMA Li et al. (2018) | 3.99 | 8.21 | 9.60 | 5.15 | 10.45 | 12.70 | 6.90 | 13.23 | 17.40 |
| | STGCN Yu et al. (2018) | 2.88 | 5.74 | 7.62 | 3.47 | 7.24 | 9.57 | 4.59 | 9.40 | 12.70 |
| | DCRNN Li et al. (2018) | 2.77 | 5.38 | 7.30 | 3.15 | 6.45 | 8.80 | 3.60 | 7.59 | 10.50 |
| | GW-Net Wu et al. (2019) | 2.69 | 5.15 | 6.90 | 3.07 | 6.22 | 8.37 | 3.53 | 7.37 | 10.01 |
| | GMAN Zheng et al. (2020) | 2.80 | 5.55 | 7.41 | 3.12 | 6.49 | 8.73 | 3.44 | 7.35 | 10.07 |
| | ASTGCN Guo et al. (2019) | 3.07 | 8.23 | 5.90 | 3.61 | 7.16 | 10.34 | 4.42 | 8.73 | 13.35 |
| | PM-MemNet Lee et al. (2022) | 2.65 | 5.29 | 7.01 | 3.03 | 6.29 | 8.42 | 3.46 | 7.29 | 9.97 |
| | MegaCRN Jiang et al. (2023b) | 2.52 | 4.94 | 6.44 | 2.93 | 6.06 | 7.96 | 3.38 | 7.23 | 9.72 |
| | PDFormer Jiang et al. (2023a) | 2.83 | 2.83 | 7.77 | 3.20 | 6.46 | 9.19 | 3.62 | 7.47 | 10.91 |
| | STD-MAE Gao et al. (2024) | 2.62 | 5.02 | 6.70 | 2.99 | 6.07 | 8.04 | 3.40 | 7.07 | 9.59 |
| | CITRUS Einizade et al. (2024) | 2.70 | 5.14 | 6.74 | 2.98 | 5.90 | 7.78 | 3.44 | 6.85 | 9.28 |
| | ModWaveMLP Sun et al. (2024) | 2.20 | 4.19 | 5.65 | 2.59 | 5.07 | 6.81 | 3.05 | 6.24 | 8.95 |
| | **SSF (Ours)** | **1.68** | **2.55** | **2.34** | **2.01** | **2.86** | **2.75** | **2.23** | **3.14** | **3.65** |
| PEMS-BAY | ARIMA Li et al. (2018) | 1.62 | 3.30 | 3.50 | 2.33 | 4.76 | 5.40 | 3.38 | 6.50 | 8.30 |
| | STGCN Yu et al. (2018) | 1.36 | 2.96 | 2.90 | 1.81 | 4.27 | 4.17 | 2.49 | 5.69 | 5.79 |
| | DCRNN Li et al. (2018) | 1.38 | 2.95 | 2.90 | 1.74 | 3.97 | 3.90 | 2.07 | 4.74 | 4.90 |
| | GW-Net Wu et al. (2019) | 1.30 | 2.74 | 2.73 | 1.63 | 3.70 | 3.67 | 1.95 | 4.52 | 4.63 |
| | GMAN Zheng et al. (2020) | 1.35 | 2.90 | 2.87 | 1.65 | 3.82 | 3.74 | 1.92 | 4.49 | 4.52 |
| | ASTGCN Guo et al. (2019) | 1.55 | 3.17 | 3.44 | 2.01 | 4.19 | 4.66 | 2.57 | 5.27 | 6.01 |
| | PM-MemNet Lee et al. (2022) | 1.34 | 2.82 | 2.81 | 1.65 | 3.76 | 3.71 | 1.95 | 4.49 | 4.54 |
| | MegaCRN Jiang et al. (2023b) | 1.28 | 2.72 | 2.67 | 1.60 | 3.68 | 3.57 | 1.88 | 4.42 | 4.41 |
| | PDFormer Jiang et al. (2023a) | 1.32 | **1.32** | 2.78 | 1.64 | 3.79 | 3.71 | 1.91 | 4.43 | 4.51 |
| | STD-MAE Gao et al. (2024) | 1.23 | 2.62 | 2.56 | 1.53 | 3.53 | 3.42 | 1.77 | 4.20 | 4.17 |
| | CITRUS Einizade et al. (2024) | 1.21 | 2.61 | 2.51 | 1.48 | 3.28 | 3.23 | 1.78 | 3.99 | 4.08 |
| | ModWaveMLP Sun et al. (2024) | 0.86 | 1.80 | 1.76 | **1.22** | 2.87 | 2.72 | 1.63 | 3.95 | 3.88 |
| | **SSF (Ours)** | **0.85** | 1.39 | **1.41** | 1.29 | **2.10** | **1.32** | 1.77 | **2.74** | **2.65** |
| NAVER-Seoul | STGCN Yu et al. (2018) | 4.63 | 6.92 | 14.49 | 5.50 | 8.83 | 17.37 | 6.77 | 10.89 | 20.42 |
| | DCRNN Li et al. (2018) | 4.86 | 7.12 | 15.35 | 5.67 | 8.80 | 18.38 | 6.40 | 10.06 | 21.09 |
| | GW-Net Wu et al. (2019) | 4.91 | 7.24 | 14.86 | 5.26 | 8.13 | 16.16 | 5.55 | 8.77 | 16.97 |
| | GMAN Zheng et al. (2020) | 5.20 | 8.32 | 16.98 | 5.35 | 8.67 | 17.47 | 5.48 | 8.94 | 17.89 |
| | ASTGCN Guo et al. (2019) | 5.09 | 7.44 | 16.14 | 5.71 | 8.73 | 18.78 | 6.22 | 9.58 | 20.37 |
| | PM-MemNet Lee et al. (2022) | 4.57 | 6.72 | 14.43 | 5.04 | 7.86 | 16.34 | 5.24 | 8.39 | 16.94 |
| | MegaCRN Jiang et al. (2023b) | 4.46 | 6.63 | 11.26 | 4.91 | 6.66 | 12.02 | 5.12 | 8.11 | 12.33 |
| | PDFormer Jiang et al. (2023a) | 4.56 | 6.74 | 14.49 | 5.01 | 7.33 | 15.69 | 5.76 | 8.91 | 17.98 |
| | STD-MAE Gao et al. (2024) | 4.20 | 5.08 | 8.32 | 4.68 | 6.13 | 11.90 | 5.20 | 7.56 | 12.10 |
| | ModWaveMLP Sun et al. (2024) | 4.01 | 5.23 | 9.11 | 4.22 | 7.61 | 13.08 | 5.32 | 8.44 | 14.10 |
| | **SSF (Ours)** | **3.41** | **5.01** | **1.03** | **3.58** | **5.84** | **1.35** | **3.84** | **7.19** | **1.63** |

projected into the sheaf spectral domain using the eigenbasis $U$ obtained from the eigendecomposition of the sheaf Laplacian.

At graph neural network layer $l$, given the hidden node features $X^{(l)}$, the following operations are applied:

$$\hat{X}^{(l)} = U^T X^{(l)} \tag{6}$$

$$\hat{X}^{(l+1)} = \hat{g}_{\text{heat}} \hat{X}^{(l)} W^{(l)} \tag{7}$$

$$X^{(l+1)} = \sigma(U \hat{X}^{(l+1)}) \tag{8}$$

where $W^{(l)}$ is a learnable weight matrix, and $\sigma(\cdot)$ denotes non-linearity. The complete SSF framework is formalized in Algorithm 1.

## 4 EXPERIMENTAL SETUP

To ensure a fair and consistent evaluation, we adopt the same split data set as established in previous works. Specifically, for the PEMS04 and PEMS08 datasets, we follow the 60:20:20 ratio of training, validation, and testing splits, as used in Song et al. (2020); Gao et al. (2024); Wang et al. (2025). For the METR-LA, PEMS-BAY, and NAVER-Seoul datasets, we employ a 70:10:20 split, in line with the protocols outlined in Lee et al. (2022); Li et al. (2018); Jiang et al. (2023b). More details on the benchmark datasets are provided in appendix A. Our experiments are conducted using PyTorch framework, running on an NVIDIA A100 GPU with 80 GB of memory. We utilize the Adam optimizer with an initial learning rate set to 0.001, a mini-batch size of 64 and MSE loss function. To mitigate overfitting, we employ early stopping based on the validation loss, with a patience threshold of 20 epochs. Graph structures for each dataset are constructed using pre-defined adjacency matrices derived from the physical distances between road segments. Each model is trained to forecast future

traffic conditions based on past temporal observations. Specifically, the input to the model consists of the previous 12 time steps (equivalent to one hour of data), and the forecasting horizon is the same (e.g., horizon 12 predicts the next 12 time steps), in case of horizon 12, same procedure applies to horizon 6 (30 minutes) and horizon 3 (15 minutes). To ensure consistent data scaling and improve convergence, we apply z-score normalization independently per sensor node across the dataset. This normalization approach is also consistent with that used in the baseline methods. Each component of our proposed framework, along with its associated hyperparameters, has been thoroughly examined through a series of controlled experiments. To understand the individual contribution and sensitivity of each element, we conduct comprehensive ablation studies, the results of which are presented in the following section and the appendices.

## 5 RESULTS AND DISCUSSION

### 5.1 QUANTITATIVE RESULTS

We present the comprehensive performance evaluation of our proposed model on the four widely-used benchmark datasets: METR-LA, PEMS-BAY Li et al. (2018), and PEMS04, PEMS08 Li et al. (2018) in Table 1 and Table 2. We compare to a wide variety of the state-of-the-art methods, details on the selected baselines are provided in appendix B. Across all datasets, our model delivers significant improvements over state-of-the-art baselines. Our method outperforms both GNN-based models (STGCN, DCRNN) and attention-driven approaches (GMAN, ASTGCN), often by substantial margins. The reason our

Table 2: Performance on PEMS04 and PEMS08 Datasets

| Model | PEMS04 | | | PEMS08 | | |
|---|---|---|---|---|---|---|
| | MAE | RMSE | MAPE | MAE | RMSE | MAPE |
| ARIMA Li et al. (2018) | 33.73 | 48.80 | 24.18 | 31.09 | 44.32 | 22.73 |
| STGCN Yu et al. (2018) | 22.70 | 35.55 | 14.59 | 18.02 | 27.83 | 11.40 |
| DCRNN Li et al. (2018) | 24.70 | 38.12 | 17.12 | 17.86 | 27.83 | 11.45 |
| GW-Net Wu et al. (2019) | 25.45 | 39.70 | 17.29 | 19.13 | 31.05 | 12.68 |
| GMAN Zheng et al. (2020) | 19.25 | 31.33 | 9.06 | 15.47 | 25.72 | 10.40 |
| ASTGCN Guo et al. (2019) | 21.80 | 32.82 | 16.56 | 16.63 | 25.27 | 13.08 |
| PM-MemNet Lee et al. (2022) | 20.63 | 33.12 | 15.22 | 19.5 | 30.13 | 12.89 |
| MegaCRN Jiang et al. (2023b) | 20.44 | 32.65 | 14.23 | 18.91 | 29.01 | 12.32 |
| PDFormer Jiang et al. (2023a) | 18.32 | 29.97 | 12.10 | 13.58 | 23.51 | 9.05 |
| STD-MAE Gao et al. (2024) | 17.80 | 29.25 | 11.97 | 13.44 | 22.47 | 8.76 |
| ModWaveMLP Sun et al. (2024) | 17.13 | 27.5 | 10.92 | 13.01 | 20.2 | 7.98 |
| SSF (Ours) | 16.76 | 34.22 | 8.22 | 9.01 | 22.16 | 4.09 |

method may slightly underperform in rare cases like PEMS04 and PEMS08 using RMSE metric, is that these datasets contain more outliers and RMSE penalizes deviations much more heavily. Most importantly, our model maintains superior accuracy across all forecasting horizons, including the challenging 60-minute horizon, where most baseline methods experience significant performance degradation.

### 5.2 QUALITATIVE RESULTS

**Robustness with challenging/fluctuating data.** While the four datasets mentioned above are considered in the literature standard benchmarks for evaluating every spatio-temporal forecasting model, we want to assess the robustness of our framework by exploring a more challenging dataset with a different environmental setting. In Table 1, we compare our framework performance to the baselines on NAVER-Seoul dataset Lee et al. (2022) which covers the entire main road network in Seoul, South Korea, and features more abrupt fluctuations in traffic speed as well as a significantly larger number of sensors (more details on the datasets are provided in the supplementary material). This makes it particularly challenging and well-suited for testing the scalability and generalization ability of spatio-temporal models. These results demonstrate the superior performance of our SSF framework compared to existing state-of-the-art approaches, showing strong error stability across horizons. Notably, the dramatic reduction in MAPE indicates that SSF captures both absolute and relative variations in the data more effectively than prior graph-based or transformer-based architectures.

**Robustness in long-term forecasting.** In figure 3 we illustrate the multi-horizon forecasting performance of SSF compared with STD-MAE and ModWaveMLP. While the competing baselines show a sharp increase in error as the horizon extends from 15 to 60 minutes, SSF demonstrates remarkable robustness, with only a modest growth in error values. The stability across both short- and long-term predictions illustrates that our sheaf-inspired framework offers a fundamentally more expressive and resilient approach to modeling long-term dependencies in complex spatio-temporal systems.

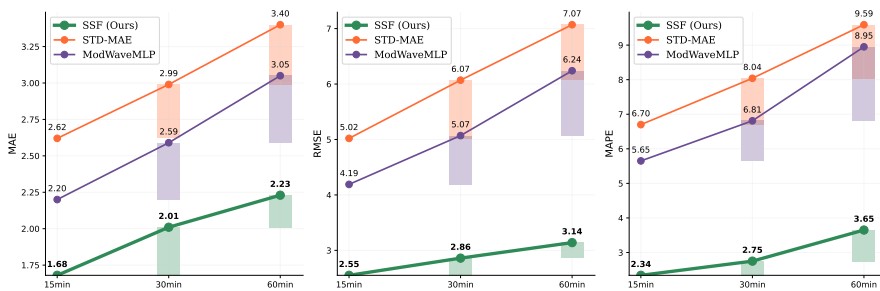

Figure 3: Comparison of forecasting performance across horizons (15, 30, 60 minutes) on METR-LA dataset. Our method achieves better incremental error compared to the baselines.

Additionally, to assess the computational impact of the proposed spectral filter, we measure the average seconds per iteration across varying sheaf stalk dimensions on NAVER-Seoul dataset.

Figure 4 demonstrates that incorporating the spectral filter consistently reduces iteration time compared to the sheaf model without the filter. While the computation of the eigendecomposition of the sheaf Laplacian introduces additional complexity, particularly in datasets with a large number of nodes and higher complexity (NAVER-Seoul), this overhead is effectively offset by the benefits of spectral filtering, where

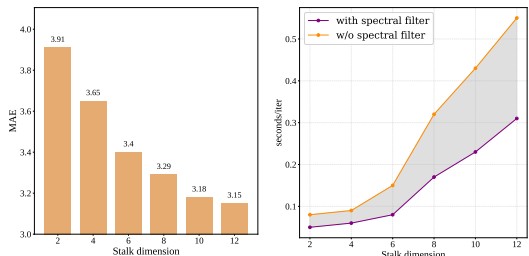

Figure 4: Impact of sheaf stalk dimension on prediction error (left), and on training time (right) with and without spectral filter.

selecting the top $k$ eigenvectors yields a more compact and expressive representation. This effect becomes more pronounced as the stalk dimension increases, highlighting the filter's efficiency in managing the growing complexity of the model.

## 5.3 ABLATION STUDIES

To understand the influence of the stalk dimensionality in our sheaf-based model, we conducted experiments on the NAVER-Seoul dataset, which is characterized by high sensor density and intricate traffic dynamics. As illustrated in Figure 4, prediction error decreases with increasing stalk dimension.

Table 3: Ablation study on the effect of spectral filtering across horizons.

| Dataset | | SSF w/o spectral filter | | | |
|---|---|---|---|---|---|
| | | 15min | 30min | 60min | Avg |
| **METR-LA** | MAE | 1.70 | 2.38 | 3.08 | 2.39 |
| | RMSE | 2.78 | 6.65 | 3.89 | 4.44 |
| | MAPE | 1.94 | 2.27 | 3.17 | 2.46 |
| **PEMS-BAY** | MAE | 1.58 | 2.02 | 1.83 | 1.81 |
| | RMSE | 2.97 | 3.19 | 3.06 | 3.07 |
| | MAPE | 1.27 | 1.38 | 3.01 | 1.89 |
| **NAVER-Seoul** | MAE | 4.55 | 5.21 | 11.6 | 7.12 |
| | RMSE | 8.64 | 9.16 | 20.72 | 12.84 |
| | MAPE | 7.02 | 11.04 | 21.44 | 13.17 |

This suggests that richer stalk representations enable the model to capture more complex local feature variations and dependencies. However, increasing the stalk dimension comes with some computational burden.

We also investigate the impact of varying the number of eigenvalues $k$ retained in the spectral filter on forecasting accuracy. As illustrated in Figure 6, increasing $k$ beyond a moderate threshold results in noticeable performance degradation across all forecasting horizons and evaluation metrics. This pattern suggests that including too many spectral components introduces noise or redundant information, thereby diminishing the discriminative power of the learned representations. In contrast, selecting a small set of dominant eigenvectors enables the model to focus on the most salient spectral features, resulting in improved accuracy and robustness.

Table 3 presents the results of an ablation study assessing the impact of removing the spectral filtering component from the proposed SSF model. Across benchmark datasets, METR-LA, PEMS-BAY, and

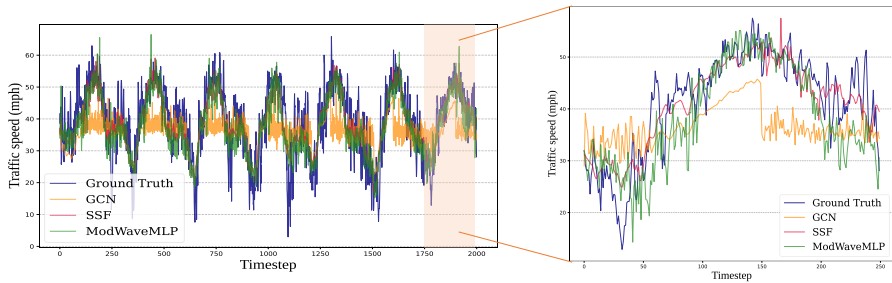

Figure 5: Comparison of traffic speed forecasting performance on NAVER-Seoul dataset.

NAVER-Seoul, performance degrades when spectral filtering is excluded. Notably, the NAVER-Seoul dataset exhibits the most pronounced drop in accuracy, particularly at longer forecasting horizons (60 minutes), where both RMSE and MAPE values significantly increase. This highlights the importance of spectral filtering in capturing essential graph frequency components that enhance spatio-temporal representation, especially in more complex or noisier traffic scenarios.

Finally, to demonstrate how our model captures the complex spatio-temporal dependencies, we visualize in Figure 5 the prediction output of SSF compared to traditional graph convolutional network (GCN) and the most competing model in the baselines (ModWaveMLP) on the challenging NAVER-Seoul dataset. SSF tends to capture the complex patterns and produce more reliable predictions.

## 6 CONCLUSION

In this work, we introduced Spectral Sheaf Filtering (SSF), a principled and robust framework for spatio-temporal representation learning that bridges cellular sheaf theory with spectral graph filtering. By redefining information flow in graph neural networks, SSF moves beyond conventional message-passing schemes, having sheaf-based structures to capture complex, higher-order dependencies through restriction maps. Our spectral formulation over the sheaf Laplacian enables a selective handling of frequency components. Extensive experiments on spatio-temporal traffic forecasting benchmarks demonstrate the superior performance of SSF, especially in challenging long-range prediction settings.

**Reproducibility Statement.** We are committed to ensuring the reproducibility of our results. To this end, detailed experimental setup is provided in section 4. Details on the public benchmark datasets are explained in Appendix A. The source code with appropriate documentation is available at: https://github.com/anonymous-submisssion/SSF.

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

SUPPLEMENTARY MATERIAL

## A  DATASETS DETAILS

We conduct our experiments on a diverse set of widely recognized real-world benchmark datasets commonly used for spatio-temporal traffic forecasting. Each dataset comprises traffic speed readings sampled at 5-minute intervals, but they differ in both spatial resolution, defined by the number of traffic sensors, and temporal duration, measured by the number of timesteps.

Firstly, we utilize the METR-LA and PEMS-BAY datasets Li et al. (2018), which were collected from freeway networks in the Los Angeles and Bay Area regions of California, respectively. These datasets have become standard benchmarks in the literature due to their rich temporal patterns and moderate spatial coverage. In addition, we include the PEMS04 and PEMS08 datasets Song et al. (2020), which originate from different regions within the Caltrans Performance Measurement System (PeMS) and provide further variability in spatial topology and traffic flow dynamics.

To further evaluate the robustness of our framework under complex urban traffic conditions, we also conduct experiments on the NAVER-Seoul dataset Lee et al. (2022). NAVER-Seoul covers the entire main road network in Seoul, South Korea, and features more abrupt fluctuations in traffic speed as well as a significantly larger number of sensors. This makes it a particularly demanding benchmark, well-suited for testing the scalability and generalization ability of spatio-temporal models. A summary of the key characteristics of all datasets used in our experiments is provided in Table 4.

Table 4: Summary of datasets' characteristics.

|               | METR-LA     | PEMS-BAY  | NAVER-Seoul | PEMS04    | PEMS08         |
|---------------|-------------|-----------|-------------|-----------|----------------|
| **Start Time**    | 03.2012     | 01.2017   | 09.2020     | 01.2018   | 07.2016        |
| **End Time**      | 06.2012     | 05.2017   | 12.2020     | 02.2018   | 08.2016        |
| **Sampling Time** | 5 minutes   | 5 minutes | 5 minutes   | 5 minutes | 5 minutes      |
| **Region**        | Los Angeles | Bay Area  | Seoul       | Bay Area  | San Bernardino |
| **Timesteps**     | 34,272      | 52,116    | 26,208      | 16992     | 17856          |
| **Spatial Units** | 207         | 325       | 774         | 307       | 170            |

## B  BASELINES

We compare our proposed model with the state-of-the-art models in spatio-temporal traffic prediction:

- Graph Convolutional Networks (GCNs): These methods use spatial graph structures to model the topology of traffic networks. We include STGCN Yu et al. (2018), a pioneering model that combines spatial graph convolutions with temporal 1D convolutions to jointly capture spatial-temporal dependencies. In addition, GW-Net Wu et al. (2019) introduces adaptive graph learning, enabling dynamic construction of the adjacency matrix to better capture latent spatial relationships.

- Recurrent Neural Networks (RNNs): These models are designed to capture temporal dependencies through sequential modeling. DCRNN Li et al. (2018) employs diffusion convolution integrated with gated recurrent units (GRUs) to model spatial-temporal dynamics on directed graphs. MegaCRN Jiang et al. (2023b) incorporates meta-graph construction and cross-node recurrent processing, improving generalization across different traffic nodes.

- Attention-based Architectures: These models utilize various attention mechanisms to selectively focus on relevant spatial and temporal features. GMAN Zheng et al. (2020) introduces a multi-level attention mechanism across both space and time, enabling flexible modeling of dynamic dependencies. ASTGCN Guo et al. (2019) employs both spatial and temporal attention modules alongside graph convolutions for fine-grained context modeling. PDFormer Jiang et al. (2023a) integrates the design of a spatial self-attention module along with graph masking matrices to capture the dynamic spatial dependencies.

- Spectral and Representation Learning Models: These models focus on frequency-domain or latent representation learning to model complex patterns. PM-MemNet Lee et al. (2022) designs a pattern

memory mechanism to enhance long-term temporal pattern retention. STD-MAE Gao et al. (2024) applies two decoupled masked autoencoders to reconstruct spatio-temporal series along the spatial and temporal dimensions. ModWaveMLP Sun et al. (2024) utilizes mode decomposition and wavelet-based denoising techniques to capture multi-scale temporal features effectively. CITRUS Einizade et al. (2024) which leverages the separability of continuous heat kernels from Cartesian graph products to perform graph spectral decomposition.

## C  SSF ALGORITHM

---

**Algorithm 1** Spectral Sheaf Filtering (SSF)

---

1: **Input:** Graph $G = (V, A)$ with $N = |V|$ nodes, input data $\mathbf{X} \in \mathbb{R}^{N \times T \times F}$, Adjacency $\mathcal{A}$, prediction horizon $H$, number of layers $L$, stalk dimension $d$
2: **Output:** Predicted signal $\hat{X}_{t+1:t+T'}$
3: **Step 1: Sheaf Construction**
4: $L_{\mathcal{F}} \leftarrow \text{sheafLaplacianConstruct}(\mathbf{X}, \mathcal{A})$
5: **Step 2: Spectral Decomposition**
6: $\tilde{L}_{\mathcal{F}} = U \Lambda U^T$             ▷ Compute eigendecomposition
7: $\lambda_1 \le \lambda_2 \le \cdots \le \lambda_{Nd}$             ▷ Sort eigenvalues
8: $k \leftarrow \min(\text{spectral\_k}, Nd)$         ▷ Number of eigenmodes to keep
9: **Step 3: Heat Kernel Filter Design**
10: **for** $i = 1$ to $Nd$ **do**
11:    $g_{heat}(\lambda_i) = e^{-\alpha \lambda_i}$
12: **end for**
13: $\hat{g}_{heat} = \text{diag}(e^{-\alpha \lambda_1}, \ldots, e^{-\alpha \lambda_{Nd}})$
14: $\tilde{\mathbf{L}}_{\text{filt}} \leftarrow \mathbf{U}_{:k} \text{diag}(\tilde{\boldsymbol{\lambda}}_{1:k}) \mathbf{U}_{:k}^T$
15: **Step 4: Spectral Filtering**
16: Initialize: $X^{(0)} \leftarrow X_t$
17: **for** layer $\ell = 0$ to $L - 1$ **do**
18:    $\hat{X}^{(\ell)} = U^T X^{(\ell)}$           ▷ Transform to spectral domain
19:    $\hat{X}_{filtered}^{(\ell)} = \hat{g}_{heat} \hat{X}^{(\ell)} W^{(\ell)}$        ▷ Apply spectral filter
20:    $\tilde{X}^{(\ell+1)} = U \hat{X}_{filtered}^{(\ell)}$       ▷ Transform back to spatial domain
21:    $X^{(\ell+1)} = \sigma(\tilde{X}^{(\ell+1)})$          ▷ Apply activation
22: **end for**
23: **Step 5: Forecasting**
24: $Z = X^{(L)}$
25: $\hat{X}_{t+\tau} = \text{MLP}_{forecast}(Z)$         ▷ Generate future predictions
26: **Return:** $\hat{X}_{t+1:t+T'} = \{\hat{X}_{t+1}, \hat{X}_{t+2}, \ldots, \hat{X}_{t+T'}\}$

---

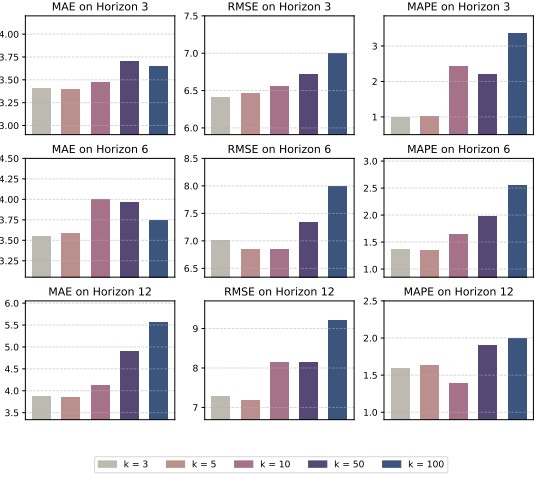

Figure 6: Effect of the number of eigenvalues $k$.

## D    EFFECT OF THE NUMBER OF EIGENVALUES (K)

Figure 6 illustrates the impact of varying the number of eigenvalues (k) used in the spectral filter on model performance across different prediction horizons. The analysis is conducted on the NAVER-Seoul dataset with five different k values: 3, 5, 10, 50, and 100. We report three evaluation metrics (MAE, RMSE, and MAPE) across three forecasting horizons (3, 6, and 12 time steps). The results reveal that using fewer eigenvalues (k=3) consistently produces the best performance across all metrics and horizons, with performance generally degrading as k increases. This trend is particularly pronounced for k=100, which shows notably worse performance. The results suggest that a lower-dimensional spectral representation (k=3) is optimal for this traffic forecasting task, indicating that the most important spectral components capture the essential spatial relationships in the traffic network, while higher-order components may introduce noise or overfitting. This finding provides valuable insights for tuning the spectral filter component of the proposed model.

## E    TIME COMPLEXITY ANALYSIS

Our method targets sparse, medium-scale road networks ($N \sim 10^2$–$10^3$, $|E| = O(N)$) with sheaf Laplacian $\mathbf{L}_F$ of size $(Nd) \times (Nd)$ and stalk dimension $d$. Constructing restriction maps and assembling $\mathbf{L}_F$ costs $O(|E|d^2) = O(Nd^2)$. Computing $k$ truncated eigenpairs ($k \ll Nd$) requires $O((Nd)^2k)$. Each layer applies dense transforms $\mathbf{U}_k^\top \mathbf{X}$ and $\mathbf{U}_k \hat{\mathbf{X}}$, costing $O(kNd^2)$. Thus, the total complexity is dominated by the eigendecomposition cost which is quadratic in $N$ for fixed $d, k$. In the following tables, we show how the forward pass time scales with varying $N, d, k$ on METR-LA dataset in Table 5, Table 6, and on synthetic data in Table 7. In practice, the quadratic term does not become a bottleneck at the scales considered for traffic networks. As we can see, even when we increase the sheaf stalk dimension to $40$ (which is not needed to have such a high value), the time remains within a favorable range, same for $N, k$. This proves that our framework remains efficient in the conditions for which it was designed.

Table 5: One forward pass time vs. stalk dimension $d$ (*fixed N=207, k=5*).

| d | 2 | 4 | 6 | 10 | 20 | 40 |
|---|---|---|---|----|----|----|
| **Forward Pass(s)** | 0.1031 | 0.1205 | 0.1706 | 0.2160 | 0.2817 | 0.4413 |

Table 6: One forward pass time vs. number of selected eigenpairs $k$ (*fixed N=207, d=6*).

| k | 3 | 5 | 10 | 20 | 50 | 100 |
|---|---|---|----|----|----|-----|
| **Forward Pass(s)** | 0.1464 | 0.1706 | 0.1974 | 0.3132 | 2.7721 | 4.3210 |

Table 7: One forward pass time vs. number of nodes in the graph $N$ (*fixed k=5, d=6*).

| N | 100 | 300 | 500 | 800 | 1000 | 1500 |
|---|-----|-----|-----|-----|------|------|
| **Forward Pass(s)** | 0.1306 | 0.1706 | 0.1773 | 0.2504 | 0.2724 | 0.3433 |

