# OpenReview forum: "Spectral Sheaf Filtering: A Topological Approach to Spatio-Temporal Modeling"
_ICLR.cc/2026/Conference — Submitted to ICLR 2026_

### Official Review · Reviewer_T1jd · 2025-10-19

**Soundness:** 2
**Presentation:** 2
**Contribution:** 2
**Rating:** 2
**Confidence:** 3

**Summary:**

This paper focuses on addressing the challenges of graph-based learning in spatio-temporal data, such as complex non-stationary dependencies and the limitations of conventional message passing. It proposes the Spectral Sheaf Filtering (SSF) framework, a pioneering approach that leverages cellular sheaf algebraic topology to redefine graph information propagation—assigning vector spaces and restriction maps to nodes and edges to capture context-dependent localized dynamics. A key innovation is the introduction of spectral filtering over the sheaf Laplacian, enabling frequency-aware decomposition via graph Fourier transform to mitigate oversmoothing in deep graph neural networks. Theoretically, the paper establishes the eigendecomposition of the sheaf Laplacian, generalizing Fourier modes to the sheaf-theoretic setting. Experimentally, on five spatio-temporal traffic forecasting benchmarks (METR-LA, PEMS-BAY, PEMS04, PEMS08, NAVER-Seoul), SSF outperforms state-of-the-art methods, especially in long-horizon forecasting, with significant error reductions and maintains efficient training speed.

**Strengths:**

1. The motivation in the introduction, especailly shown in Figure 1 is convincing and easy to catch. The perspective of utilizing sheaf representation for spatial-temporal graph tasks is novel.
2. The formulation of sheaf representation and analysis is very clear.
3. The paper conducts comprehensive experiments to evaluate the proposed methods, the the prediction accuracy improves significantly.

**Weaknesses:**

1. The most severe concern is the clarity of the proposed model. Though the introduction of sheaf formulation is clear, it is unclear how the spatail-temporal information of **previous t timestamps** of node is utilized to output the prediction of nodes in **the further T' timestamps**.  It is also unclear what is the loss function and how the model is trained. This concern is crucial that affects the clarity of this paper.
2. The paper lacks the complexity analysis for the proposed method, as the Laplacian decomposition is time-consuming. It is also better to list the running time of decomposition in appendix F in addition to the training runtime.

**Questions:**

See details in weakenss.

---

> ### Author Response · Authors · 2025-11-26
>
> We thank the reviewer for recognizing the novelty of the paper and for mentioning several strengths, and for their constructive feedback. We now addressed the mentioned weaknesses and included them in the improved manuscript. We would appreciate a re-evaluation.
>
> ### Clarity of data processing
> For a graph with $N$ nodes, the model receives its historical observations over the past $T$ time steps, where each time step contains $F$ features (traffic speed), resulting in an input tensor of shape $(N, T \times F)$. These past $T$ temporal frames are concatenated into a single feature vector per node. So, time is treated as part of the input feature dimension. This representation is then passed through an initial MLP that projects it into a higher-dimensional embedding space of size $d \cdot h$, where $d$ is the sheaf stalk dimension and $h$ is the hidden layer size. The spatial dependencies are captured through the sheaf Laplacian. Then at each layer, the model performs a graph Fourier transform with respect to the eigenvectors of the sheaf Laplacian, applies a heat-kernel, and transforms the signal back to the spatial domain.
> The final node representations are collapsed back and mapped through a linear layer to construct the output predictions of shape $(N, T' \cdot F)$, which are the forecasts for the next $T'$ future time steps. Training is performed using the MSE loss between the predicted and ground-truth future observations.
>
> We now added this clarification of how the model processes the data into the manuscript, and an improved diagram of data flow will be added to the camera-ready version.
>
> ### Complexity Analysis
> Our method targets sparse, medium-scale road networks (typically $N \sim 10^{2}$-$10^{3}$, $|E| = O(N)$) with sheaf Laplacian $\mathbf{L}\_{F}$ of size $(Nd) \times (Nd)$ and stalk dimension $d$.
> Constructing restriction maps and assembling $\mathbf{L}\_{F}$ costs $O(|E|d^{2}) = O(N d^{2})$. Computing $k$ truncated eigenpairs ($k \ll Nd$) requires $O((Nd)^{2}k)$. Each layer applies dense transforms $\mathbf{U}\_{k}^{\top}\mathbf{X}$ and
> $\mathbf{U}\_{k}\,\hat{\mathbf{X}}$, costing $O(k N d^{2})$.
> Thus, the total complexity is dominated by the
> eigendecomposition cost which is quadratic in $N$ for fixed $d, k$.
> In the following tables, we show how the forward pass time scales with varying $N, d, k$ on METR-LA dataset in the first two tables, and on synthetic data in the third table. In practice, the quadratic term does not become a bottleneck at the scales considered for traffic networks, putting also in consideration that in real-world data, traffic networks are mainly sparse not fully-connected graphs. As we can see, even when we increase the sheaf stalk dimension to $40$ (which is not needed to have such a high value), the time remains within a favorable range, same for $N, k$. This proves that our framework remains efficient in the conditions for which it was designed.
>
> (fixed N = 207, k = 5)
> | **d** | 2 | 4 | 6 | 10 | 20 | 40 |
> |-------|---|----|----|-----|------|------|
> | **Forward Pass (s)** | 0.1031 | 0.1205 | 0.1706 | 0.2160 | 0.2817 | 0.4413 |
>
> (fixed N = 207, d = 6)
> | **k** | 3 | 5 | 10 | 20 | 50 | 100 |
> |-------|----|----|------|------|-------|--------|
> | **Forward Pass (s)** | 0.1464 | 0.1706 | 0.1974 | 0.3132 | 2.7721 | 4.3210 |
>
> (fixed k = 5, d = 6)
> | **N** | 100 | 300 | 500 | 800 | 1000 | 1500 |
> |-------|------|------|------|------|--------|--------|
> | **Forward Pass (s)** | 0.1306 | 0.1706 | 0.1773 | 0.2504 | 0.2724 | 0.3433 |
>
> For the model on METR-LA, with N=207, k=5, d=6, the avg forward time is 0.17s, and the avg backprop time is 0.0031s. The total avg time for 1 epoch is 13.91s.
> The eigendecomposition of the sheaf Laplacian is performed once per forward pass and costs on avg : 0.04 seconds.
>
> We included these clarifications in the manuscript and made a lot of improvements according to your valuable feedback, if there are any other questions or concerns, we will be happy to answer them during discussion period.

---

> ### Author Response · Authors · 2025-11-28
>
> Dear reviewer,
>
> We thank you for taking the time to review our manuscript and for your insightful comments. As the end of discussion period is approaching, we would like to discuss any remaining concerns or points of confusion. Your feedback is highly valuable for strengthening the manuscript, and we look forward to your response.
>
> Sincerely,
>
> Authors

---

### Official Review · Reviewer_KfCh · 2025-10-25

**Soundness:** 3
**Presentation:** 3
**Contribution:** 3
**Rating:** 6
**Confidence:** 4

**Summary:**

The paper introduces *Spectral Sheaf Filtering (SSF)* for spatio temporal node regression/prediction task. It builds a cellular sheaf with learned restriction maps on vertex–edge incidences, forms the *sheaf Laplacian*, and performs propagation via spectral heat kernel filtering. A rolling window of past graph signals is processed slice by slice through these sheaf spectral layers, followed by a small MLP head for multi horizon node prediction. Experiments on five traffic datasets report strong performance with ablations on filtering, stalk dimension, and number of modes.

**Strengths:**

1. **Clean spatial operator.** The cellular sheaf with learned restriction maps yields a well-defined sheaf Laplacian and an interpretable quadratic form that directly encodes edge-wise compatibility.

2. **Principled frequency control.** Spectral heat kernel filtering in the sheaf eigenbasis addresses oversmoothing while retaining useful high frequency content; ablations on filter on or off, stalk dimension (d), and top (k) modes support the design.

3. **Strong empirical performance.** Competitive results across standard traffic benchmarks, clear algorithmic presentation, and interpretability hooks via low frequency modes and the nullspace.

**Weaknesses:**

1. **Temporal head underpowered.** After a strong spatial backbone, the temporal modeling is a small MLP. A comparison to lightweight temporal modules (1D temporal conv, GRU, tiny attention) is missing.

2. **Metric sanity.** Some reported MAPE values look unusually small given the MAE and RMSE, suggesting possible differences in definitions, units, or de normalization. Baseline training and metric computation parity are not fully documented.

3. **Domain breadth.** All experiments are traffic speed forecasting. A second non traffic domain (or transfer tests) would strengthen claims of general utility.

4. **Efficiency specifics.** Top (k) eigensolvers can dominate runtime. The schedule for recomputing eigenpairs and a timing breakdown are not fully reported.

**Questions:**

1. **Temporal modeling.** Please compare your MLP head to a small temporal module per node: 1D temporal conv, GRU, or a tiny attention block. Report accuracy and wall clock.

2. **Metrics.**  Several tables show MAPE notably lower than what the reported MAE would imply. Please provide exact formulas for MAE, RMSE, and MAPE, confirm that evaluation is on de normalized predictions, state the units, and specify any epsilon or clipping used in the MAPE denominator.

3. **Scalability.** How often are eigenpairs recomputed during training. What are typical (k) and stalk dimension (d). Include a timing breakdown for eigensolve, forward, and backward, and a scaling curve with (N), (d), and (k).

4. **Generality.** Can you add one non traffic dataset of the same task type (e.g., environmental, power, or something similar) or a generalization test: inductive node split or cross city transfer.

---

> ### Author Response · Authors · 2025-11-27
>
> We thank the reviewer for the positive assessment and recognizing several strengths of the paper and for their constructive comments. We now addressed all the mentioned concerns.
>
> ### Q1:
>
> Our model architecture performs global spectral filtering with heat kernels at every layer. The spectral filter acts jointly on the spatial and temporal dimensions. Thus, the temporal structure is already captured in the spectral backbone, not all the burden is put on the final projection layer. This is also consistent with prior ST-GNN work where the final temporal head becomes small when there is a sufficiently expressive backbone.
> We now performed experiments with replacing our MLP layer with 1D temporal convolution, and one temporal attention layer.
>
> | **Dataset**     | **Horizon** | **Temporal Head** | **RMSE** | **MAE** | **MAPE** |
> |-----------------|-------------|-------------------|----------|---------|----------|
> | **METR-LA**     | 3           | Temporal Conv     | 4.20     | 2.10    | 1.77     |
> |                 |             | Temporal Attn     | 2.52     | 1.43    | 1.57     |
> |                 |             | MLP               | 2.55     | 1.68    | 2.34     |
> |                 | 6           | Temporal Conv     | 4.93     | 3.23    | 3.31     |
> |                 |             | Temporal Attn     | 2.89     | 1.96    | 2.33     |
> |                 |             | MLP               | 2.86     | 2.01    | 2.75     |
> |                 | 12          | Temporal Conv     | 5.94     | 4.04    | 5.75     |
> |                 |             | Temporal Attn     | 3.67     | 2.87    | 3.25     |
> |                 |             | MLP               | 3.14     | 2.23    | 3.65     |
> | **NAVER-Seoul** | 3           | Temporal Conv     | 7.49     | 4.86    | 5.80     |
> |                 |             | Temporal Attn     | 11.99    | 7.36    | 6.07     |
> |                 |             | MLP               | 5.01     | 3.41    | 1.03     |
> |                 | 6           | Temporal Conv     | 9.49     | 11.63   | 10.15    |
> |                 |             | Temporal Attn     | 8.85     | 4.96    | 3.08     |
> |                 |             | MLP               | 5.84     | 3.58    | 1.35     |
> |                 | 12          | Temporal Conv     | 11.62    | 13.43   | 11.78    |
> |                 |             | Temporal Attn     | 11.29    | 6.33    | 3.47     |
> |                 |             | MLP               | 7.19     | 3.84    | 1.63     |
>
>
> The training time is:
> | **Temporal Head**       | **Avg Forward(s)** | **Avg Backprop(s)** | **One Epoch(s)** |
> |-----------------|--------------------|----------------------|-------------------|
> | MLP             | 0.1706             | 0.0031               | 13.91             |
> | Temporal Conv   | 0.2112             | 0.0081               | 17.87             |
> | Temporal Attn   | 0.3048             | 0.0109               | 25.15             |
>
> While the temporal attention improves short horizons on the smallest dataset, it doesn't keep its stability over long horizons or for the bigger dataset. The MLP layer keeps the stable performance with being 2.8× faster than the temporal attention layer.
>
> ### Q2:
> All reported metrics are computed on denormalized predictions. Thus, all errors are measured in the native units of the datasets (mph), an example of model predictions is shown in Figure $5$. The exact formulas we use:
>
> $$
> \mathrm{MAE} = \frac{1}{NT'} \sum\_{i=1}^{N} \sum\_{t=1}^{T'}
> \left| \hat{Y}\_{i,t} - Y\_{i,t} \right|
> $$
>
> $$
> \mathrm{RMSE} = \sqrt{\frac{1}{NT'} \sum\_{i=1}^{N} \sum\_{t=1}^{T'}
> \left(\hat{Y}\_{i,t} - Y\_{i,t}\right)^{2}}
> $$
>
> $$
> \mathrm{MAPE} = \frac{1}{|\\{(i,t): Y\_{i,t} \neq 0\\}|\} \sum\_{\substack{i,t \\ Y\_{i,t} \neq 0}}
> \left| \frac{\hat{Y}\_{i,t} - Y\_{i,t}}{Y\_{i,t}} \right|
> $$
>
> The denominator of MAPE excludes only zero-valued ground-truth entries, exactly as in other baseline implementations.
> When the absolute prediction error is $1$-$2$ mph, the corresponding MAPE naturally falls into the $1$--$3\%$ range for traffic speed values. Thus, the reported MAPE values are mathematically consistent. We think the reasons why some baselines have dramatical increase of error in longer horizons or fluctuating data like NAVER-Seoul, is because of the model instability. In contrast, our model remains very stable. Also, to confirm the error calculations, we trained a model on $1$ epoch only then tested on the unseen test set, the MAPE value was $11.12$ while the model trained on $80$ epochs got MAPE error of $1.32$ on the test set. So, the error decreases with model learning. We also added pretrained models in the GitHub repo for ensuring result reproduction.

---

> ### Author Response · Authors · 2025-11-27
>
> ### Q3:
>
> **Complexity Analysis:** Our method targets sparse, medium-scale road networks (typically $N \sim 10^{2}$-$10^{3}$, $|E| = O(N)$) with sheaf Laplacian $\mathbf{L}\_{F}$ of size $(Nd) \times (Nd)$ and stalk dimension $d$.
> Constructing restriction maps and assembling $\mathbf{L}\_{F}$ costs $O(|E|d^{2}) = O(N d^{2})$. Computing $k$ truncated eigenpairs ($k \ll Nd$) requires $O((Nd)^{2}k)$. Each layer applies dense transforms $\mathbf{U}\_{k}^{\top}\mathbf{X}$ and
> $\mathbf{U}\_{k}\,\hat{\mathbf{X}}$, costing $O(k N d^{2})$.
> Thus, the total complexity is dominated by the
> eigendecomposition cost which is quadratic in $N$ for fixed $d, k$.
> In the following tables, we show how the forward pass time scales with varying $N, d, k$ on METR-LA dataset in the first two tables, and on synthetic data in the third table. In practice, the quadratic term does not become a bottleneck at the scales considered for traffic networks, putting also in consideration that in real-world data, traffic networks are mainly sparse not fully-connected graphs. As we can see, even when we increase the sheaf stalk dimension to $40$ (which is not needed to have such a high value), the time remains within a favorable range, same for $N, k$. This proves that our framework remains efficient in the conditions for which it was designed.
>
> (fixed N = 207, k = 5)
> | **d** | 2 | 4 | 6 | 10 | 20 | 40 |
> |-------|---|----|----|-----|------|------|
> | **Forward Pass (s)** | 0.1031 | 0.1205 | 0.1706 | 0.2160 | 0.2817 | 0.4413 |
>
> (fixed N = 207, d = 6)
> | **k** | 3 | 5 | 10 | 20 | 50 | 100 |
> |-------|----|----|------|------|-------|--------|
> | **Forward Pass (s)** | 0.1464 | 0.1706 | 0.1974 | 0.3132 | 2.7721 | 4.3210 |
>
> (fixed k = 5, d = 6)
> | **N** | 100 | 300 | 500 | 800 | 1000 | 1500 |
> |-------|------|------|------|------|--------|--------|
> | **Forward Pass (s)** | 0.1306 | 0.1706 | 0.1773 | 0.2504 | 0.2724 | 0.3433 |
>
> For the model on METR-LA, with N=207, k=5, d=6, the avg forward time is 0.17s, and the avg backprop time is 0.0031s. The total avg time for 1 epoch is 13.91s.
> The eigendecomposition of the sheaf Laplacian is performed once per forward pass and costs on avg : 0.04 seconds.
>
> ### Q4:
>
> Thank you for this suggestion. we conducted a node-inductive generalization test, where we trained the model on a subset of nodes and evaluated it on completely unseen nodes. As expected, the inductive node split leads to performance degradation compared to the original SSF pipeline. However, the model still outperforms strong baselines under this more challenging setting, and this is a direct proof of the model's ability to generalize the learned patterns.
>
> | **Data**        | **15min / horizon 3 MAE** | **RMSE** | **MAPE** | **30min / horizon 6 MAE** | **RMSE** | **MAPE** | **60min / horizon 12 MAE** | **RMSE** | **MAPE** |
> |-----------------|---------------------------|----------|----------|-----------------------------|----------|----------|------------------------------|----------|----------|
> | **METR-LA**     | 2.20                      | 3.10     | 2.40     | 2.51                        | 3.67     | 3.90     | 2.66                         | 4.02     | 3.95     |
> | **PEMS-BAY**    | 1.22                      | 1.95     | 2.05     | 1.42                        | 2.55     | 2.65     | 1.84                         | 3.25     | 2.70     |
> | **NAVER-Seoul** | 3.85                      | 5.40     | 1.78     | 4.10                        | 5.40     | 2.32     | 4.55                         | 7.90     | 2.52     |
>
>
> We sincerely thank the reviewer for these valuable comments which strengthened our paper, we've edited the manuscript accordingly, and the rest will be added to the camera-ready version. If there are any other questions or concerns, we will be happy to answer them during the rebuttal phase.

---

> ### Comment · Reviewer_KfCh · 2025-11-27
>
> Thank you for the detailed rebuttal and additional experiments. Your responses largely address most of my concerns about the temporal head, metric definitions, and computational complexity, and the new node-inductive evaluation is a helpful addition within the traffic domain. However, the core novelty and evidence around oversmoothing/interpretability and sheaf vs standard spectral baselines are still somewhat incremental and specialized. Also, the empirical scope remains focused entirely on traffic forecasting without a second non-traffic domain, so the broader claim of general spatio-temporal applicability is not fully demonstrated. I also find it hard to verify exactly which rebuttal changes have already been incorporated into the current PDF, since revisions are not visibly marked (maybe a different color). Overall, I am positively updated by the rebuttal but will keep my score at 6.

---

### Official Review · Reviewer_Ph9G · 2025-10-31

**Soundness:** 1
**Presentation:** 2
**Contribution:** 1
**Rating:** 0
**Confidence:** 5

**Summary:**

The paper proposes Spectral Sheaf Filtering (SSF) for spatio-temporal forecasting. The authors (i) build a cellular sheaf on a sensor graph with learnable restriction maps, (ii) define a sheaf Laplacian and apply spectral filtering via a heat kernel, performing “message passing in the spectral domain”, and (iii) report results on benchmark datasets. Key claims include improved expressivity, mitigation of over-smoothing, global receptive fields with O(1) operations, and reduced computational burden from using spectral filtering.

**Strengths:**

- Clear motivation to move beyond uniform edge propagation; sheaves are a natural tool to model asymmetric/high-order relations.

**Weaknesses:**

- The core technical component, spectral filtering of a Laplacian, is classical in graph signal processing (GSP), including design/learning of filters and universal approximation with FIR/IIR (e.g., Shuman et al., 2013; Sandryhaila & Moura, 2013). The paper does not acknowledge this body of work adequately nor contrast SSF against modern spatio-temporal spectral approaches (e.g., Einizade et al., NeurIPS 2024, which provides stability/over-smoothing analysis and SOTA forecasting). As written, replacing the standard Laplacian by a sheaf Laplacian plus a heat kernel filter is an incremental variant of well-established spectral filtering; the “novelty” claim (Sec. 1, contributions) is overstated.
- The paper claims spectral methods “achieve a global receptive field in O(1) operations”, computing (and back-propagating through) eigendecompositions is not O(1) and is typically O(kN^2) for k eigenvector-eigenvalue pairs. Therefore, the manuscript claims spectral filtering reduces computational burden without a rigorous cost analysis. These are contradictory/incorrect as stated.
- “This decomposition is interpretable” is claimed but not demonstrated.
- Oversmoothing and expressivity improvements are claimed but not theoretically analyzed, and the empirical evidence directly targeting these phenomena is missing.
- The scalability claim in the conclusion is not substantiated. While Table 6 reports per-epoch training time, scalability is dominated by (i) the eigendecomposition of the (sheaf) Laplacian, and (ii) dense transforms like $U^T X$. As the number of nodes increases, both the eigenvalue decomposition and the dense multiplications become bottlenecks in time and memory.
- The paper states a spectral decomposition $L_F=U \Lambda U^T$ and interprets eigenpairs as “frequencies”, but never establishes conditions under which the proposed Laplacian is symmetric positive semidefinite (and thus diagonalizable by an orthonormal basis with nonnegative eigenvalues). This is critical for using a heat kernel and a Fourier-like transform.
- Equations (6-8) implement project-filter-reconstruct (Fourier transform, diagonal filter, inverse transform). Calling this a message passing layer is misleading: it is spectral filtering.
- No comparison against sheaf GNNs trained in the spatial domain. Without this, it is unclear whether the gain comes from the sheaf modeling or simply from spectral filtering.
- No comparison with CITRUS (Einizade et al., NeurIPS 2024), which reports SOTA on the same class of tasks on longer horizons, and includes oversmoothing analysis. Similarly, no comparison with more classical ARMA graph-temporal filters.
- The algorithm input mentions “hyperedge index” although the core model is defined on graphs.
- The claim “novel spectral filtering” is too broad. Spectral filtering for graph signals is a decade-old area (Shuman et al., 2013; Sandryhaila & Moura, 2013; many follow-ups), with universal filter design and time-graph IIR filters; recent spatio-temporal spectral GNNs exist. Please moderate claims and cite appropriately.


**General comment**: The paper reads like it was written heavily relying on LLMs: a repetitive introduction, broad claims stated without proof, and numerous inconsistencies. Even if LLMs were used, which is acceptable, the authors have not verified facts, moderate claims, and aligned the narrative with the actual technical content.

**Questions:**

- Under which assumptions on $L_F$ is symmetric PSD?
- What is the actual computational complexity?
- Is $\alpha$ in the heat kernel a learned parameter per layer or a fixed hyperparameter? How is it selected?

---

> ### Author Response · Authors · 2025-11-23
>
> We appreciate the reviewer's critical thoughts. Here, we address all their concerns and questions, and hope for a reconsideration.
>
> ### W1, W11:
> The core novelty of this work originally appears in the new way of learning representation of spatio-temporal data, although sheaves are a powerful mathematical formulation to model high-order relations, they’re very underrepresented in the literature of spatio-temporal work, same as investigating the effect of spectral methods on the sheaf Laplacian.
>
> A standard graph Laplacian encodes only adjacency-based coupling. However, the sheaf Laplacian operator $L_{F}$ satisfies:
>
> $$
> (L_{F}x)_{v}=\sum\_{e=(u,v)} F\_{v\triangleleft e}^{\top} F\_{v\triangleleft e} x\_{v} - F\_{v\triangleleft e}^{\top}F\_{u\triangleleft e} x\_{u}
> $$
>
>
> which means feature-dependent info is encoded. This fundamentally changes the geometry of diffusion and spectral modes.
>
> Introducing a way of learning representation that results in good performance gains is the main interest of ICLR community.
>
> Thank you for pointing out these fundamental references, we agree that this body of work should be properly acknowledged in the scope of our paper. We amended the manuscript accordingly.

---

> ### Author Response · Authors · 2025-11-23
>
> ### W2, W5, Q2, Q3:
> We would like to answer all complexity-related questions here and we added it to the manuscript.
> Firstly, to clarify, computing eigendecompositions is definitely not $\mathcal{O}(1)$. But while spectral filtering introduces the cost of eigendecomposition, with a small/mid-size graph, some spectral methods can achieve high efficiency in terms of number of parameters, as highlighted in (Bruna et al., 2014). In our experimentaion (with small $k$ and moderate $N,d$), using spectral filtering empirically reduces training time compared to spatial SheafGNN propagation (as shown in Fig. 4), even though it requires an eigendecomposition. This effect becomes more pronounced as the stalk dimension increases.
> The intended meaning was: once the eigenbasis is computed, each layer can achieve global mixing by performing matrices transformation on the whole graph. In addition, the heat kernel uses a fixed parameter $\alpha$, selected empirically in our implementation. Low-order or localized polynomial spectral kernels correspond to $K$-hop localized filters, while the heat kernel can expand the effective receptive field as $\alpha$ increases.
>
> We edited and clarified now this part in the manuscript.
>
> ### Complexity Analysis
> Our method targets sparse, medium-scale road networks (typically $N \sim 10^{2}$-$10^{3}$, $|E| = O(N)$) with sheaf Laplacian $\mathbf{L}\_{F}$ of size $(Nd) \times (Nd)$ and stalk dimension $d$.
> Constructing restriction maps and assembling $\mathbf{L}\_{F}$ costs $O(|E|d^{2}) = O(N d^{2})$. Computing $k$ truncated eigenpairs ($k \ll Nd$) requires $O((Nd)^{2}k)$. Each layer applies dense transforms $\mathbf{U}\_{k}^{\top}\mathbf{X}$ and
> $\mathbf{U}\_{k}\,\hat{\mathbf{X}}$, costing $O(k N d^{2})$.
> Thus, the total complexity is dominated by the
> eigendecomposition cost which is quadratic in $N$ for fixed $d, k$.
> In the following tables, we show how the forward pass time scales with varying $N, d, k$ on METR-LA dataset in the first two tables, and on synthetic data in the third table. In practice, the quadratic term does not become a bottleneck at the scales considered for traffic networks, putting also in consideration that in real-world data, traffic networks are mainly sparse not fully-connected graphs. As we can see, even when we increase the sheaf stalk dimension to $40$ (which is not needed to have such a high value), the time remains within a favorable range, same for $N, k$. This proves that our framework remains efficient in the conditions for which it was designed.  We do not claim scalability to arbitrary massive graphs.
>
> (fixed N = 207, k = 5)
> | **d** | 2 | 4 | 6 | 10 | 20 | 40 |
> |-------|---|----|----|-----|------|------|
> | **Forward Pass (s)** | 0.1031 | 0.1205 | 0.1706 | 0.2160 | 0.2817 | 0.4413 |
>
> (fixed N = 207, d = 6)
> | **k** | 3 | 5 | 10 | 20 | 50 | 100 |
> |-------|----|----|------|------|-------|--------|
> | **Forward Pass (s)** | 0.1464 | 0.1706 | 0.1974 | 0.3132 | 2.7721 | 4.3210 |
>
> (fixed k = 5, d = 6)
> | **N** | 100 | 300 | 500 | 800 | 1000 | 1500 |
> |-------|------|------|------|------|--------|--------|
> | **Forward Pass (s)** | 0.1306 | 0.1706 | 0.1773 | 0.2504 | 0.2724 | 0.3433 |
>
>
> ### References
> Bruna, J., Zaremba, W., Szlam, A. and LeCun, Y., 2013. Spectral networks and locally connected networks on graphs. arXiv preprint arXiv:1312.6203.

---

> ### Author Response · Authors · 2025-11-23
>
> ### W3, W6, Q1:
>
> The interpretability in the manuscript refers to the fact that the eigenvectors of the sheaf Laplacian $L\_{\mathcal{F}}$ form orthonormal frequency modes when it is shown to be symmetric positive semidefinite exactly as in classical GSP. The sheaf Laplacian $L_{\mathcal{F}}$ is symmetric positive semidefinite when:
> the base graph is undirected, edge stalks have inner-product structures, and all restriction maps satisfy $F\_{v \triangleleft e}^{\top} F\_{v \triangleleft e} \in \mathcal{S}\_d^{+}$. Also, incidence contributions are accumulated symmetrically.
>
> These conditions are already satisfied by our parameterization
> (all $(F\_{v \triangleleft e}$) are real matrices with symmetric block
> contributions).
> Thus, the interpretibility claim is not speculative, the dominant modes retained by the filter correspond to low-frequency, coherent traffic patterns. We stated the conditions now explicitly in the paper.

---

> ### Author Response · Authors · 2025-11-23
>
> ### W4:
>
> Several previous works show that using the classical graph Laplacian, which implies minimizing the classical Dirichlet energy function
> $\sum\_{u,v} A\_{uv} (x\_u - x\_v)^2=x^\top L x$
> encourages neighboring nodes to take similar values and associate this behaviour with oversmoothing, such as in [3, 4, 5]. Instead, our sheaf-based formulation minimizes implicitly a more expressive energy function with learned restriction maps:
> $$
> E^{\mathcal{F}}\_{L\_2}(x) = \frac{1}{2} \sum\_{e} \left\lVert F\_{v \triangleleft e} D\_v^{-\tfrac{1}{2}} x\_v - F\_{u \triangleleft e} D\_u^{-\tfrac{1}{2}} x\_u \right\rVert\_2^2
> $$
> So, it basically penalizes
> $(F\_{v\triangleleft e}  x\_v - F\_{u\triangleleft e}  x\_u)^{2}$
> rather than $(x\_v - x\_u)\^{2}$. The restriction maps ( $F\_{v\triangleleft e} $) are learned ( $d \times d $) matrices that break uniform aggregation.
> Thus, we can argue that it has a direct effect on reducing oversmoothing.
>
> ### W7:
> In principal, equations (6–8) are exactly spectral filtering. However, the term "message passing" in GNN literature has evolved to denote the block of an individual graph layer, regardless of whether the implementation is in the spatial or spectral domain, that it aggregates and transforms information across the graph. In [1, 2], it was explicitly stated that both spatial and spectral GNNs are in fact MPNN.
> However, for more clarity, we now removed the “message passing” term.
>
> ### W8:
> Comparison with SheafGNN trained in the spatial domain was already included in our ablation studies Section 5.3, Table 3. Spatial sheafGNN performs worse than full SSF framework in general. The performance gap is largest on Naver-Seoul dataset with the most irregular traffic patterns.
>
> ### W9:
> Thank you for the suggestion, we agree that comparing to these methods will be insightful, thus we added the comparison in the results section.
>
> ### W10:
> That was unintentional typo, because we're trying to build a uniform framework that can be extended to hypergraphs in future work, so we adopted this notation in our implementation. But the scope of this work is regular graphs, so we've edited that in the manuscript.
>
>
> ### General Comment
> We would like to clarify that LLMs were never used in this paper for ideation, theoretical formulation, full paragraph writing, or code generation at any stage of this work. All claims mentioned in the paper were supported by experimental evidence on real-world standard benchmarks and the code repo is publicly available with step by step documentation to ensure reproducibility.
>
> We've made many changes to the manuscript according to your valuable feedback and we added references to the fundamental related work. If there are any further concerns or questions, we'll be happy to answer them.
>
> ### References
> 1. Balcilar, M., Renton, G., Héroux, P., Gaüzère, B., Adam, S. and Honeine, P., 2021. Analyzing the expressive power of graph neural networks in a spectral perspective. In International conference on learning representations.
>
> 2. Balcilar, M., Héroux, P., Gauzere, B., Vasseur, P., Adam, S. and Honeine, P., 2021, July. Breaking the limits of message passing graph neural networks. In International Conference on Machine Learning (pp. 599-608). PMLR.
>
> 3. Shuman, D.I., Narang, S.K., Frossard, P., Ortega, A. and Vandergheynst, P., 2013. The emerging field of signal processing on graphs: Extending high-dimensional data analysis to networks and other irregular domains. IEEE signal processing magazine, 30(3), pp.83-98.
>
> 4. Einizade, A., Malliaros, F. and Giraldo, J.H., 2024. Continuous product graph neural networks. Advances in Neural Information Processing Systems, 37, pp.90226-90252.
>
> 5. Duta, I., Cassarà, G., Silvestri, F. and Liò, P., 2023. Sheaf hypergraph networks. Advances in Neural Information Processing Systems, 36, pp.12087-12099.

---

> > ### Comment · Reviewer_Ph9G · 2025-11-27
> >
> > Thank you for the updates, weaknesses W1, W2, W5, W7, W9, W10, and W11 are satisfactorily addressed. The overall quality of the paper has increased.
> >
> > I still have the following concerns regarding the revised manuscript and rebuttal:
> >
> > **W3**: The claim that eigendecomposition directly leads to interpretability remains unconvincing.
> >
> > **W4**: While the oversmoothing argument is acknowledged, the claim requires theoretical or empirical support.
> >
> > **W6**: For the stated conditions under which the sheaf Laplacian is positive semidefinite: This still requires either a proper citation or a complete proof.
> >
> > **W8**: The comparison in Table 3 reflects the presence or absence of spectral filtering (*i.e.*, heat kernel application), not a comparison against other sheaf-based methods. As a result, the experiment does not isolate the contribution of the sheaf component itself, which remains a crucial missing evaluation.
> >
> > Given these remaining issues, I update my score to 2.

---

> > > ### Author Response · Authors · 2025-11-27
> > >
> > > We would like to thank you for your feedback and remaining engaged during discussion.
> > >
> > > In the paper, we already cite the fundamental work introduced by (Bodnar et al., 2023). We also reference them in our code, where we reused some of the sheaf building blocks from their twitter-research repo. In their paper, they prove the PSD conditions of the sheaf Laplacian, so it wouldn't be necessary that we rewrite the proofs in our paper.
> > >
> > > Regarding the effect of the sheaf structure itself: Table 3 already isolates the contribution of the sheaf component, because disabling spectral filtering directly reduces our model to a pure spatial SheafGNN. In Figure 4, we also provide isolated experiment on the effect of sheaf dimension.
> > >
> > > We agree that oversmoothing analysis should be added to strengthen the paper, we will add it to the camera-ready version due to the limited time window of rebuttal.
> > >
> > > We remain confident that our proposed approach adds value to spatio-temporal modeling. And we would be happy to discuss more any points of confusion during the rebuttal phase.

---

### Official Review · Reviewer_nsJs · 2025-11-03

**Soundness:** 4
**Presentation:** 2
**Contribution:** 3
**Rating:** 8
**Confidence:** 2

**Summary:**

The paper applies a new sheaf-based graph network approach for spatiotemporal forecasting. The model achieves very impressive results on the METR-LA dataset.

**Strengths:**

- Originality: The proposed method SSF is novel in traffic prediction.
- Quality: The model is validated on several traffic datasets. The data preprocessing pipelines seem consistent with SOTA methods.
- Clarity: The paper provides appropriate background for readers to understand the sheaf GNNs.
- Significance: The reported improvement on METR-LA is significant.

**Weaknesses:**

- Clarity
  - The paper could benefit from a better description of the information flow. While Figure 2 shows the general framework, it is unclear how spatiotemporal data is processed by SSF.
  - Most works in the field report MAE, RMSE, MAPE on all 3 horizons (3, 6, 12). Having Table 5 in the main paper instead of Table 1 would allow for better comparison.
- Motivation
  - It seems like sheaf GNNs were designed and evaluated on hypergraph datasets. Traffic networks are static. The paper would benefit from showing the need for sheaf GNNs in traffic data.

**Questions:**

- Is the temporal dimension of the data flattened and used as the node embedding? Is that what line 18 does in Algorithm 1? Is there a linear layer to project the initial node embeddings to the hidden space?
- In the provided GitHub codebase, you call `python run.py --model_id metr_12`. Do you train a separate model for each forecasting horizon?
- Can you provide a pretrained model for result reproduction? Are the baseline method results copied from the original papers?

---

> ### Author Response · Authors · 2025-11-23
>
> We thank the reviewer for the positive assessment and appreciate the recognition of paper novelty.
>
> ### W1, Q1:
> For a graph with $N$ nodes, the model receives its historical observations over the past $T$ time steps, where each time step contains $F$ features (traffic speed), resulting in an input tensor of shape $(N, T \times F)$. These past $T$ temporal frames are concatenated into a single feature vector per node. So, time is treated as part of the input feature dimension. This representation is then passed through an initial MLP that projects it into a higher-dimensional embedding space of size $d \cdot h$, where $d$ is the sheaf stalk dimension and $h$ is the hidden layer size. The spatial dependencies are captured through the sheaf Laplacian. Then at each layer, the model performs a graph Fourier transform with respect to the eigenvectors of the sheaf Laplacian, applies a heat-kernel, and transforms the signal back to the spatial domain.
> The final node representations are collapsed back and mapped through a linear layer to construct the output predictions of shape $(N, T' \cdot F)$, which are the forecasts for the next $T'$ future time steps. Training is performed using the MSE loss between the predicted and ground-truth future observations. We now added this clarification of how the model processes the data into the manuscript, and an improved diagram of data flow will be added to the camera-ready version.
>
> ### W2:
> Thank you for this suggestion, we now moved Table 5 to the main text.
>
> ### W3:
> While traffic graphs are static, their relational structure is highly heterogeneous and asymmetric (as illustrated in the example at the introduction). In standard GCNs, every edge uses the same propagation rule, which is known to collapse the heterogeneous dependencies and induce oversmoothing. Sheaves introduce learnable restriction maps to enable richer representation.
> Our ablations (Fig. 4) show that increasing the sheaf stalk dimension $d$ consistently improves accuracy, directly proving that traffic forecasting benefits from the richer geometric representation of sheaves.
>
> ### Q2:
> model_id in the code is just a naming for the output format, can be named anything, we train the same model architecture for all tasks.
>
> ### Q3:
> We now released pretrained models trained on different datasets in the GitHub repo under "pretrained_models" folder. In general, and for fair comparison, we run all experiments for baselines on all tasks (which were also consistent with the results reported in their papers), except in some individual cases where their model implementation had some issues and we couldn't run their code, so we reported the results directly from their original papers.
>
> If there are any other questions or concerns, we will be happy to answer them during the rebuttal phase.

---

### Author Response · Authors · 2025-11-30

We would like to extend our gratitude to all reviewers for their time and constructive feedback. We are glad the reviewers recognized the value of our work. More specifically:

**Originality**: "The proposed method SSF is novel in traffic prediction" **(reviewer nsJs)**, "The perspective of utilizing sheaf representation for spatial-temporal graph tasks is novel" **(reviewer T1jd)**.

**Clarity and Motivation**: "Clear motivation to move beyond uniform edge propagation" **(reviewer Ph9G)**, "The paper provides appropriate background for readers to understand the sheaf GNNs" **(reviewer nsJs)**, "The motivation in the introduction, especailly shown in Figure 1 is convincing and easy to catch, the formulation of sheaf representation and analysis is very clear." **(reviewer T1jd)**, "clear algorithmic presentation" **(reviewer KfCh)**.

**Experiments and Results**: "The model is validated on several traffic datasets. The data preprocessing pipelines seem consistent with SOTA methods. The reported improvement on METR-LA is significant." **(reviewer nsJs)**, "Strong empirical performance. Competitive results across standard traffic benchmarks" **(reviewer KfCh)**, "The paper conducts comprehensive experiments to evaluate the proposed methods, the the prediction accuracy improves significantly" **(reviewer T1jd)**.

We also would like to highlight that **we addressed all the reviewers concerns and comments in the rebuttal**, particularly:

- We added comprehensive complexity analysis for our complete framework.
- We added detailed experiments on the effect of changing the parameters $N, d, k$ individually with respect to the training time.
- We improved the method diagram and included detailed description of how data is processed at each stage of the model.
- We included the full results on all horizons in the main text.
- We added pretrained models in the GitHub repo for results reproduction.
- We improved the introduction, clarified the contributions.
- We referenced the established fundamental related works on spectral filtering.
- We added comparisons with other spectral based methods.
- We did experiments on the effect of changing the temporal head.

---

### Meta-Review · Area_Chair_4NUB · 2026-01-06

**Summary:**

There are several major concerns:
1. The manuscript is poorly written. The reviewers pointed out that the manuscript lacks clarity in  data processing steps, training procedures, and metrics definitions, leading to confusion about MAPE consistency and baseline comparability.
2. There are also concerns in the novelty of this work. The core spectral filtering technique is classical in graph signal processing; reviewers questioned the novelty and requested clearer comparisons to recent spatio-temporal spectral methods (e.g., CITRUS, ARMA-based filters) to isolate the specific contribution of the sheaf framework.
3. After the rebuttal, the concerns theoretical justification remain unaddressed. Key theoretical aspects were seen as inadequately justified, including conditions for the sheaf Laplacian being symmetric PSD (for spectral decomposition validity), explicit analysis supporting oversmoothing mitigation, and interpretability derived directly from eigendecomposition.
4. For the experiments, they focus solely on traffic datasets, raising doubts about general spatio-temporal applicability. Reviewers also noted insufficient ablations clearly separating the gains attributable specifically to sheaf modeling versus spectral filtering.

**Reviewer Concerns:**

The concern regarding to method complexity have been incorporated. Some of the descriptions of the work are further clarified by the authors. However, the major concerns are still outstanding.

**Reviewer Scores:**

Reviewer nsJs  rated 8 with very short review without much information in both strengths and weaknesses. The reviewer are likely to maintain his rating.

Reviewer Ph9G rated 0 initially, and raised a bunch of questions. Authors  addressed some critical points (novelty positioning, computational analysis) but left unresolved interpretability, oversmoothing theory, and isolated sheaf contributions. As stated by Ph9G, it is reasonable to raise to score to 2.

Reviewer KfCh rated 6 initially. While authors addressed several points, major concerns regarding novelty and generality remained partially unresolved. Thus, Reviewer KfCh would likely maintain their rating at 6.

Reviewer T1jd rated 2 initially, mainly citing clarity and complexity concerns. Although clarity improved significantly, fundamental issues around novelty and empirical scope remained. Hence, Reviewer T1jd would likely maintain their rating at 2.

---

### Decision · Program_Chairs · 2026-01-26

Reject